# Gene Expression of Putative Pathogenicity-Related Genes in *Verticillium dahliae* in Response to Elicitation with Potato Extracts and during Infection Using Quantitative Real-Time PCR

**DOI:** 10.3390/pathogens10050510

**Published:** 2021-04-23

**Authors:** Xiaohan Zhu, Arbia Arfaoui, Mohammad Sayari, Lorne R. Adam, Fouad Daayf

**Affiliations:** Department of Plant Science, University of Manitoba, 222 Agriculture Building, Winnipeg, MB R3T2N2, Canada; huhu772@hotmail.com (X.Z.); arbia_arfaoui2002@yahoo.fr (A.A.); Mohammad.Sayari@umanitoba.ca (M.S.); Lorne.Adam@umr.umanitoba.ca (L.R.A.)

**Keywords:** *Verticillium dahliae*, effectors, defense inducers, highly aggressive, gene expression

## Abstract

Quantitative real-time PCR was used to monitor the expression of 15 *Verticillium dahliae*’s genes, putatively involved in pathogenicity, highly (HAV) and weakly aggressive (WAV) *V. dahliae* isolates after either (i) elicitation with potato leaf, stem, or root extracts, or (ii) inoculation of potato detached petioles. These genes, i.e., coding for Ras-GAP-like protein, serine/threonine protein kinase, Ubiquitin-conjugating enzyme variant-MMS2, NADH-ubiquinone oxidoreductase, Thioredoxin, Pyruvate dehydrogenase E1 VdPDHB, myo-inositol 2-dehydrogenase, and HAD-superfamily hydrolase, showed differential upregulation in the HAV versus WAV isolate in response to plant extracts or after inoculation of potato leaf petioles. This suggests their potential involvement in the observed differential aggressiveness between isolates. However, other genes like glucan endo-1,3-alpha-glucosidase and nuc-1 negative regulatory protein *VdPREG* showed higher activity in the WAV than in the HAV in response to potato extracts and/or during infection. This, in contrast, may suggest a role in their lower aggressiveness. These findings, along with future functional analysis of selected genes, will contribute to improving our understanding of *V. dahliae*’s pathogenesis. For example, expression of *VdPREG* negatively regulates phosphorus-acquisition enzymes, which may indicate a lower phosphorus acquisition activity in the WAV. Therefore, integrating the knowledge about the activity of both genes enhancing pathogenicity and those restraining it will provide a guild line for further functional characterization of the most critical genes, thus driving new ideas towards better Verticillium wilt management.

## 1. Introduction

Verticillium wilt, caused by *V. dahliae* or *V. albo-atrum*, is a major disease that engenders significant yield losses in potato production, valued at millions of dollars each year [1]. Additionally, referred to as potato early dying (PED), this disease can cause 5 to 12 metric tons of tuber yield loss per hectare, and result in up to 90% disease severity [1,2,3,4]. Previous research showed a correlation between the density of *V. albo-atrum* in the soil and severity of wilt symptoms under favorable conditions [5]. *V. dahliae* has a larger host range than *V. albo-atrum* [1], affecting more than 200 dicotyledonous plant species including flowers, oilseed and fiber crops, fruits, vegetables, and woody perennials [1,4].

*V. dahliae* produces microsclerotia that can be released in the soil and remain viable for 10 to 15 years [4,6]. Microsclerotia germinate in response to stimulation by root exudates secreted in the rhizosphere [7] and later reach the plant roots [8]. *V. dahliae* can enter the susceptible plants from the root tip, then extend to cross the endodermis and penetrate the young xylem elements, finally reaching the vascular cylinder [9]. *V. dahliae* sporulates in the vascular tissue and moves following the sap stream, but may be trapped in pit cavities [8]. Their conidia germinate in the vessel end walls and penetrate the adjacent elements for future propagation in the host [8]. 

Plants have evolved a series of defense mechanisms against *V. dahliae* [10]. Some plant species such as tomato, pea, and cotton, can compromise the expansion of *V. dahliae* by increasing the production of lignin-like phenolic polymers or lignin depositions in cortical plant cell walls [11,12]. In tomato, *Ve*-mediated plant resistance to *V. dahliae* involves the accumulation of lignin [13], which can decrease microsclerotia’s viability even in crop residues in the soil [14]. Additionally, plants accumulate several enzymes, including peroxidase, phenylalanine ammonia-lyase (PAL), and cinnamyl alcohol dehydrogenase, which are involved in plant resistance to *V. dahliae* [11,13]. Similarly, cotton plant cells reinforce their walls by the accumulation of polysaccharides (callose and cellulose) and release phytoalexins (i.e., coumarins, terpenoids) in response to infection by *V. dahliae* [15,16,17]. When *V. dahliae* enters the xylem vessels, plant vascular cell walls may produce coating materials such as cellulose to prevent fungal horizontal spread. In addition, the neighboring parenchyma cells produce paramural deposition such as pectin, callose, and cellulose [18].

Commonly used strategies to manage Verticillium wilt in crops include cultural practices such as crop rotations, and sometimes the use of fumigants [19]. However, none of these methods provide full control of the disease. Crop rotation of up to 5 years does not suppress the disease effectively [20] due to the long-lasting survival of microsclerotia in soil and the wide host range of *V. dahliae* [1,6]. Furthermore, once *V. dahliae* enters the plant vascular tissues, no fungicide were shown to control it [8]. 

Understanding the molecular mechanism that make *V. dahliae* a strong pathogen, including the identification of genes that are critical for infection [21], would set the ground for the development of better alternative approaches to control the wilt disease it causes. A previous proteomics study showed differential accumulation of a set of proteins in highly aggressive versus weakly aggressive isolates of *V. dahliae* [22]. Another study demonstrated differential gene expression in a highly versus weakly aggressive isolates in response to potato root extracts [23]. One of these genes, an isochorismatase hydrolase, was later found to be involved in *V. dahliae*’s interference with potato’s SA defense pathway [24]. The differentially expressed proteins Thioredoxin (VdTRX) (VDAG_04529), NADH-ubiquinone oxidoreductase (VDAG_09026), Pyruvate dehydrogenase E1 component subunit beta (VdPDHB) (VDAG_01642), Ubiquitin-conjugating enzyme variant MMS2 (VDAG_05365), HAD-superfamily hydrolase (VDAG_08490), Serine 3-dehydrogenase (VDAG_09532), and Wos2 (VDAG_08865), were visible only in the highly aggressive isolate’s profile [22]. Genes encoding Ras-GAP like protein (VDAG_01012), Xanthine dehydrogenase (VDAG_07735), myo-inositol 2-dehydrogenase (VDAG_08205), and DNA-(apurinic or apyrimidinic site) lyase (DNA AP lyase) (VDAG_02445), were also upregulated in the highly aggressive isolate in response to root extracts from both susceptible and moderately resistant potato cultivars [23]. Genes such as those coding for serine/threonine protein kinase (VDAG_04632), glucan endo-1,3-alpha-glucosidase agn1 (VDAG_04101), DNA repair protein RAD51 (VDAG_08796), and nuc-1 negative regulatory protein preg (PREG) (VDAG_06766) were upregulated in both highly and weakly aggressive isolates in response to potato root extracts [23]. The two transcriptomic and proteomic studies by El-Bebany et al. [22,23] clearly showed the differential ability of highly versus weakly aggressive isolates to express genes and/or proteins both constitutively in vitro, or as a response to potato root extracts. We chose 15 most prominent of those genes and proteins for further investigation in the current study, as they previously showed involvement in host-pathogen interaction. For example, Agn1 codes for an important enzyme during cytokinesis in fission-yeast. It is delivered to the interface between the cell wall and out layer of primary septum, and then participates in cell separation [25,26,27,28,29]. A ubiquitin binding site of Yeast Ubiquitin E2 Variant (UEV) Protein MMS2 is essential for DNA-damage tolerance controlled by the RAD6 pathway [30]. Serine 3-dehydrogenase is involved in the catabolism of serine. A serine 3-dehydrogenase in mammalian fungi pathogen *Paracoccidioides lutzii* interacts with host macrophages and may play roles in colonization and spread into the host [31,32]. Myo-inositol dehydrogenase is essential for inositol and inositol phosphate metabolism in aerobacter, the ability of nitrogen fixation and nodulation for host in *Sinorhizobium fredii*, and rhizopine utilization in *S. meliloti* [33,34,35]. Ras-GAP protein and serine threonine protein kinase are involved in close-related signaling transduction, controlling cell differentiation and proliferation. Ras protein regulates signaling transduction in cell growth and differentiation in both mammals and fungi. Ras-GAP protein inactivates Ras protein by accelerating the conversion from GTP-bound Ras active state to GDP–bound inactive state [36,37]. Serine/threonine protein kinases play important roles in signal transduction for fungal development and proliferation as well as in morphogenesis changes during infection [38]. The Ras family proteins can be activated by serine/threonine protein kinase [39]. Moreover, Ras protein can also control serine/threonine protein kinase to regulate the pathogenicity in various fungi [40,41,42]. Wos2 is important for cell proliferation in the yeast *S. pombe* [43]. Pyruvate dehydrogenase E1 component subunit beta is required for full assembly of the Pyruvate dehydrogenase multienzyme complex (PDHc), while full functionality of this complex is critical in catalyzing pyruvate into acetyl-CoA [44,45,46,47]. Xanthine dehydrogenase catalyzes xanthine to uric acid and also participated in purines and pyrimidines oxidation in various organisms including chicken and *Arabidopsis thaliana* [48,49,50,51,52].

As the next step, the objective of this study was to assess the expression pattern of these most prominent 15 genes, clustered in different groups according to their functions (i) in response to elicitation with different potato extracts, and (ii) during infection of the petiole of detached potato leaves by each of two *V. dahliae* isolates with contrasting levels of aggressiveness. For this purpose, and to mimic as much as possible what happens in nature, we first assessed the gene expression in *V. dahliae* in response to potato plant exudates; then, we investigated their expression during infection of potato detached leaves. Thus far, no studies have assessed the expression of these genes with respect to understanding their potential involvement in pathogenicity of *V. dahliae*. In addition, this study provides an integrated overview of how the most prominent genes express during two important phases of their lifecycles: induction by plant exudates and actual tissue infection. 

## 2. Results

### 2.1. Isolate Aggressiveness and Disease Assessment

The highly aggressive isolate (Vd1396-9) induced significantly more disease measured as total AUDPC of “percentage of infection” and “disease severity” than the weakly aggressive isolate (Vs06-07) or water control (Figure 1A,B). The plant growth rate of potato inoculated with Vd1396-9 was significantly lower than that of the other two treatments (Figure 1C). Potato plants inoculated with Vd1396-9 exhibited more symptoms at 5 weeks after inoculation (WAI) than plants inoculated with Vs06-07 or water (Figure 1D). The percentage of infection of plants inoculated with the weakly aggressive isolate was only slightly higher than in the water control, but significantly lower than with the highly aggressive one (Figure 2A). The plants inoculated with the weakly aggressive isolate showed similar disease severity and plant growth rates as water controls (Figure 2B–D).

### 2.2. Differential Fungal Gene Expression in Response to Elicitation with Potato Extracts and Inoculation

All of the 15 genes/proteins used in this research were processed for prediction for fungal signal peptides and effectors by machine-learning with SignalP 5.0 (http://www.cbs.dtu.dk/services/SignalP/ (accessed on 7 March 2021)) [54] and EffectorP (http://effectorp.csiro.au/ (accessed on 7 March 2021)) [55,56]. Glucan endo-1,3-alpha-glucosidase agn1 was predicted as a secretory protein with a probability of 0.9129 (Appendix A and Appendix A). Ubiquitin-conjugating enzyme variant MMS2, Serine 3-dehydrogenase, and myo-inositol 2-dehydrogenase were predicted as fungal effectors (Appendix A).

Most of the selected investigated genes in the current study showed higher expression in the highly aggressive isolate during infection of the detached leaves. However, the same genes exhibited different expression patterns depending on elicitation of potato root, stem, or leaf extracts. In this study, gene expression with fold change ≥ 2 were defined as up-regulated, and fold change ≤ 0.5 were defined as down-regulated.

#### 2.2.1. Genes Involved in Cell Differentiation and Proliferation

Four genes involved in cell differentiation and proliferation were assessed: Ras protein, serine/threonine protein kinase, Wos2, and glucan endo-1,3-alpha-glucosidase. Most of these genes were upregulated (fold change ≥ 2) in both highly and weakly aggressive isolates of *V. dahliae* in response to elicitation with different potato extracts (Figure 2A,C,E,G). The expression of both Ras-GAP and serine/threonine protein kinase was more noticeable in the highly than the weakly aggressive isolate in response to the stem extracts (Figure 2A,C). 

During infection of detached potato leaves, both tested isolates had increases in transcripts of Ras-GAP, serine/threonine protein kinase, and endo-1,3-alpha-glucosidase (Figure 2B,D,F). 

The accumulation of Ras-GAP transcripts was significantly higher in the highly aggressive isolate than the weakly aggressive one (Figure 2B). This was evident throughout the entire monitored period, but especially so at the early stages of infection (Figure 2B). Serine/threonine protein kinase gene reached a maximum expression in the highly aggressive isolate from 5 to 8 DAI, which was also significantly higher than the weakly aggressive one at 8 DAI (Figure 2D). The expression of endo-1, 3-alpha-glucosidase in the weakly aggressive isolate peaked at 5DAI and was higher than in the highly aggressive isolate during infection of detached leaves. In the same experiment, the expression of Wos2 was downregulated in both isolates (fold change ≤ 0.5) (Figure 2F,H).

#### 2.2.2. Genes Involved in DNA Repair

We assessed the expression of three genes involved in DNA damage repair of post-replicative processes. DNA-(apurinic or apyrimidinic site) lyase, ubiquitin-conjugating enzyme variant MMS2, and DNA repair protein RAD51 had been well investigated in yeast and mammals [57,58,59,60,61].

In most cases in our study, these three genes were up regulated in one or both isolates in response to potato extracts (Figure 3A,C,E). The expression of DNA-(apurinic or apyrimidinic site) lyase and RAD51 coding genes were higher in the weakly aggressive isolate following exposure to the potato root extract (Figure 3A,E). RAD51-coding gene, also exhibited the same response to leaf extract (Figure 3E). We also noted increased expression of RAD51-coding gene in the pre-infection stage in the weakly aggressive isolate (Figure 3E). During infection of detached leaves, only the DNA-(apurinic or apyrimidinic site) lyase-coding gene showed an increase (fold change ≥ 2) in both *V. dahliae* isolates (Figure 3B). The expression of this gene also increased with root extract elicitation, suggesting its involvement in both pre-infection and infection stages. The expression of the MMS2-coding gene in the weakly aggressive isolate during infection was suppressed (fold change ≤ 0.5) and significantly lower than in the highly aggressive isolate (Figure 3D). The expression of RAD51-coding gene was opposite to MMS2, in that its expression in the highly aggressive isolate was lower than in the weakly one at 1 and 8 DAI (Figure 3F). 

#### 2.2.3. Genes Related to ROS Production and Cleavage

Two genes related to ROS balance, thioredoxin (Trx) and NADPH ubiquinone oxidoreductase-coding genes, were assessed. Both isolates increased their expression of Trx in response to potato root extract elicitation and during detached leaf infection (Figure 4A,B). However, during detached leaves’ infection, in the highly aggressive isolate, Trx reached a dramatically high level of expression at 5 DAI and then fell back down at 8 DAI (Figure 4B). Overall, Trx expression was still higher than in the weakly aggressive isolate (Figure 4B). The expression of NADH-ubiquinone oxidoreductase-coding gene increased in the highly aggressive isolate in response to all three forms of potato extracts, though only the stem extract was statistically significant (Figure 4C). During detached leaf infection, its expression in the highly aggressive isolate was only slightly increased at 8 DAI, whereas its expression in the weakly aggressive isolate was significantly lower than in the highly aggressive isolate from 5 DAI (Figure 4D). 

#### 2.2.4. Genes Related to Cellular Metabolism 

Cellular metabolism includes many chemical reactions involved in maintaining normal life for an organism. The expression of Serine 3-dehydrogenase, pyruvate dehydrogenase E1 component subunit beta, xanthine dehydrogenase, myo-inositol dehydrogenase, and nuc-1 negative regulatory protein VdPREG coding genes, all involved in cell metabolism, were assessed. These genes are involved in catabolism of serine, conversion of pyruvate to Acetyl-CoA, oxidative metabolism of purines and pyrimidines, inositol, and inositol phosphate metabolism, and the regulation of the activity of transcription factor NUC-1, respectively [31,35,44,46,48,51,62,63].

In general, the expression of Serine 3-dehydrogenase, pyruvate dehydrogenase E1 component subunit beta and xanthine dehydrogenase increased in both *V. dahliae* isolates in response to potato extracts (Figure 5A,C,E). During infection, the expression of Serine 3-dehydrogenase and xanthine dehydrogenase was significantly up-regulated over time in the highly aggressive *V. dahliae* isolate. However, the expression of these two genes in the weakly aggressive isolate either decreased or did not change (Figure 5B,F). In addition, the highly aggressive isolate exhibited more transcripts of serine 3-dehydrogenase in response to root extracts (Figure 5A). This was also true for the highly aggressive isolate’s expression during infection of detached leaves (Figure 5B). Serine 3-dehydrogenase, pyruvate dehydrogenase E1 component subunit beta, and xanthine dehydrogenase all exhibited higher activities in the highly than the weakly aggressive *V. dahliae* isolate (Figure 5A–F).

The expression level of myo-inositol dehydrogenase-coding gene in the highly aggressive isolate was up-regulated in response to potato extracts and during infection of detached leaves, but we were unable to detect its expression in the weakly aggressive isolate in any of the treatments (Figure 5G,H). The expression of nuc-1 negative regulatory protein VdPREG in the weakly aggressive *V. dahliae* isolate was significantly higher than in the highly aggressive one under all treatments, as well as during infection at 8 DAI (Figure 5I,J).

#### 2.2.5. Genes Related to Detoxification

HAD superfamily hydrolase constitutes the largest branch of phosphatases superfamily [64,65]. In *Escherichia coli* and *S. cerevisiae*, several homologues of HAD-like hydrolase coding gene were identified [66,67]. These homologues have a role in the detoxification of 2-deoxyglucose or phosphorylated glycerol phosphates and carbohydrates [66,67].

The expression of HAD in *V. dahliae* was not significantly different between the two isolates in response to potato extracts (Figure 6A). However, during the detached leaves’ infection, HAD superfamily hydrolase transcripts accumulated more in the highly aggressive isolate (Figure 6B). Following infection, there was an increasing versus decreasing expression trends in the highly versus weakly aggressive isolates, respectively (Figure 6B).

## 3. Discussion

*V. dahliae* causes one of the most devastating potato diseases, leading to great losses worldwide [1]. Comprehending the molecular mechanisms of infection, colonization, and tissue damage are important tools towards the development of sustainable management of this disease in the field [1,4]. This study provides more comparative details about the quantitative expression of 15 genes involved in different cellular processes in two *V. dahliae* isolates possessing differential levels of aggressiveness (highly aggressive Vd1396-9 versus weakly aggressive Vs06-07). The 15 genes included in the current study showed differential expression when comparing a highly and a weakly aggressive isolates, and in response to potato root extracts [23]. In order to provide as complete a picture as possible, we assessed the expression of these 15 *V. dahliae* genes in two different settings that would relate to two different phases of *V. dahliae*’s interaction with potato in nature. Firstly, our assessment of gene expression in *V. dahliae* in response to plant exudates mimics the pre-infection phase when the pathogen senses the presence of the host roots. Secondly, assessment of gene expression in detached petiole/leaves mimics a phase in the host infection process.

Ras-GAP-like protein, serine/threonine protein kinase, Wos2, and glucan endo-1,3-alpha-glucosidase are involved in regulating cell differentiation and proliferation. The increased activity of the above mentioned genes during exposure to potato extracts, or during infection process, shows that these genes may be involved in spore germination and other cellular processes leading to infection by *V. dahliae*. This result is in line with previous findings in *Candida albicans* (behaves as a pathogenic yeast in human under specific conditions), which showed transient increases in Rho1-GAP (Rho1 belongs to Ras family) and WOS2 during a morphogenesis switch from a less-virulent yeast to a more-virulent filamentous form [68]. Ras-GAP’s involvement parallels findings that Ras-GAP is important for cell differentiation, such as yeast in bud formation of *Saccharomyces cerevisiae* [69] and hyphal growth, conidiation, and mitosis of *Aspergillus nidulans* [70]. Moreover, Rac1, another member of Ras superfamily, cooperates with p21-activated kinase Cla4, controlling the location of ROS, then in turn impacting the polarized growth and virulence of *V. dahlia* [71].

The similar expression trends of Ras-GAP and serine/threonine protein kinase during the detached leaves’ infection mirror those from the stem extract treatment, which suggests that Ras-GAP and serine/threonine protein kinase may work in a similar model, or even in the same signal pathway, in *V. dahliae*. It has been shown in numerous studies that Ras-GAP-related Ras is essential for virulence or infection-related structure formation in *F. graminearum*, *M. oryzae*, and *A. fumigatus* [40,41,42], and its function involves serine/threonine protein kinases such as MAP kinase and sometimes protein kinase A (PKA) [72,73]. This is in support of our finding that Ras-GAP and serine/threonine protein kinase express more in response to elicitation with stem extracts and infection of detached leaves by the highly aggressive isolate than the weakly aggressive one. Meanwhile, a PKA catalytic subunit gene *VdPKAC1* was found to be important for conidia and microsclerotia production, as well as pathogenicity in *V. dahliae* [74]. All of this indicates that the small GTPase signaling pathway in *V. dahliae*, including Ras-GAP and serine/threonine protein kinase, may have an important role in the infection and morphogenesis processes related to pathogenicity, and may also play important roles in differential aggressiveness between isolates. 

In *M. oryzae*, disruption of transcription factor Tup1 induced a loss of pathogenicity and an increase in glucan endo-1, 3-alpha-glucosidase’s (MoAgn1) expression [75]. This indicates that a high activity of MoAgn1, controlled by the pathogenicity-related transcription factor Tup1, may be associated with low pathogenicity. Interestingly, a β-1,6-endoglucanase gene, *VegB*, involved in degrading chitin and β-glucan, was also identified in *V. dahliae*. Single gene disruption did not significantly affect virulence compared to wild-type strains [76]. However, double-disruption mutants for both *VegB* and *VdPKAC1* in *V. dahliae* showed higher virulence than the single mutant *VdPKAC1* [76], suggesting that VegB may negatively regulate the PKA-related signaling pathway’s role in pathogenicity. In our work, the expression of endo-1, 3-alpha-glucosidase in *V. dahliae* also exhibited a transient increase in the weakly aggressive isolate. We hypothesized that the more aggressive *V. dahliae* isolate may have down-regulated the activity of endo-1, 3-alpha-glucosidase to accommodate the stress conditions in place during infection of the host.

In response to pathogen attacks, plants produce a great number of ROS that can induce DNA damage and restrict the expansion of pathogens [77]. In *S. cerevisiae*, homologues of RAD6 and RAD18 are involved in both the error-free post-replication DNA repair and the REV3-mediated mutagenesis pathways [78,79,80,81,82,83,84]. MMS2 is only involved in the error-free post-replication DNA repair mode [78]. All three genes involved in DNA repair showed an increased expression in response to potato extracts. Based on our results, DNA repair proteins could help *V. dahliae* to recover from DNA damage. This is in line with a RAD14 transient induction during transition from a less virulent to a more virulent state in *C. albicans* [68]. However, RAD51 showed a higher activity in weakly aggressive isolate than in highly aggressive one. We speculated that this protein might be more sensitive in lower aggressive isolates in response to DNA damage caused by plant defense mechanisms. In *Helicobacter pylori*, the disruption of a homologue of DNA-(apurinic or apyrimidinic site) lyase increased sensitivity to oxidative stress and reduced the colonization rate as compared to the wild type [79,85]. This indicates that DNA-(apurinic or apyrimidinic site) lyase is involved in aiding bacteria’s survival under stress conditions and plays a role in its ability to colonize the host [85]. This is similar to our findings and indicates that this homologue in both isolates may help *V. dahliae* survive stress conditions, including DNA damage during infection. Ubiquitin-conjugating enzyme variant MMS2 transcripts contribute more in the highly aggressive isolate during the infection process. This is also similar to the findings on the human pathogen *Paracoccidioides brasiliensis*, where MMS2 was significantly elevated while infecting oral keratinocytes cell [86], indicating its function during infection.

In fungi, ROS are critical for the control of infection morphogenesis, sexual development, ascospore germination, and pathogenicity [87,88,89]. An NADPH oxidase for ROS production, VdNoxB, and a tetraspanin VdPls1, are both important for the transcription factor VdCrz1 activation towards penetration peg formation, then in controlling *V. dahlia*’s virulence [90]. The Non-enzymatic ROS are produced in the mitochondria electron transport chain [91,92]. NADH-ubiquinone oxidoreductase (complex I) is the predominant gene responsible for the production of ROS in mitochondria of mammalian and bacterial cells [93,94,95,96]. Homeostasis of cellular ROS is maintained by scavenging systems which include the thioredoxin (Trx) system that consists mainly of Trx, Trx reductase (TrxR), and peroxidase (Prx) [97]. Our findings based on Trx and NADH-ubiquinone oxidoreductase in *V. dahliae* show that the ROS balance in *V. dahliae* responds differently in these two isolates and when exposed to different parts of the potato plant. The dramatic increase of TRX in the highly aggressive *V. dahliae* isolate, compared to the weakly aggressive one, suggests that the Trx function in reducing ROS and maintaining ROS balance in *V. dahliae* may be important for infection-related morphology and differentiation and therefore critical during infection. Previous studies showed that the expression of Trx or TrxR in *Staphylococcus aureus*, *S. cerevisiae*, *Schizosaccharomyces pomb*, *C. albicans*, and *Cryptococcus neoformans*, increased in response to various stress conditions such as oxidative or nitrosative stress [98,99,100,101,102,103,104]. Together, these findings suggest that the thioredoxin (Trx) ROS scavenging system may help the highly aggressive *V. dahliae* to respond to oxidative stress conditions imposed by plant defenses during colonization or at the time of infection. ROS produced by the NADH-ubiquinone oxidoreductase-mediated non-enzymatic system may also be involved in regulating spore germination during the pre-infection process. Therefore, ROS regulation may be important for pathogenicity in *V. dahliae* and possibly for the differential fitness of the highly aggressive isolate.

The genes involved in cellular metabolism are essential for maintaining nutrition support and normal growth in various organisms [31,33,34,35,44,45,46,47,48,49,50,51,62,63,105]. In *V. dahliae*, a MADS-box transcription factor VdMcm1 is critical for conidia and microsclerotia’s formation, as well as cell-wall integrity and virulence, and its deletion also affects secondary metabolism in *V. dahliae* [106]. An ubiquitin ligase (E3) enzyme controlling secondary and lipid metabolism, named VdBre1, was identified in *V. dahliae* and is important for its virulence on cotton [107]. Another *V. dahliae* bZIP transcription factor Atf1 was also essential for virulence, and this protein controlled nitrogen metabolism in both *V. dahliae* and *F. graminearum* [108]. In our study, the expression of Serine 3-dehydrogenase, pyruvate dehydrogenase E1 component subunit beta, xanthine dehydrogenase, and myo-inositol dehydrogenase were upregulated in *V. dahliae* with potato extracts elicitation and during infection. VdPREG is a negative regulator for phosphorus acquisition, suggesting that the phosphorus acquisition pathway may be more activated in the highly than in the weakly aggressive *V. dahliae* isolate. 

Several other studies show that these components of cellular metabolism are also important for virulence. D-serine metabolism is essential for virulence and regulation of several virulence factors in *Staphylococcus saprophyticus* [109]. This supports our results in *V. dahliae* that Serine 3-dehydrogenase has more activity in the highly aggressive isolate. The importance of pyruvate dehydrogenase E1 component subunit beta in the highly aggressive *V. dahliae* isolate during the pre-infection and infection stages may be suggested by its higher expression in response to stem extracts and during infection (Figure 5C,D). This is in line with the important role of Lipoamide dehydrogenase (Lpd), a member of the pyruvate dehydrogenase complex family, in the pathogenesis of *Mycobacterium tuberculosis* [110]. Plants produce antifungal compounds such as flavonoids to inhibit the activity of the pathogen’s xanthine dehydrogenase in order to increase resistance to fungi and bacteria [52,111,112,113,114]. Even though there is no direct evidence showing that xanthine dehydrogenase is essential for pathogenicity in fungi, our results showed higher activity of xanthine dehydrogenase in the highly aggressive isolate than the weakly aggressive one (Figure 5E,F). This may indicate that xanthine dehydrogenase in a highly aggressive *V. dahliae* may overcome these inhibitors from plants and further play a role in colonization. 

Our finding on myo-inositol dehydrogenase (Figure 5G,H) also suggests that the highly aggressive isolate utilizes this protein during the pre-infection and infection processes. This gene may play a function in helping *V. dahliae* in rhizopine utilization and viability for better colonization in the plant root, as previously shown in *S. meliloti* and *S. fredii* [33,34,35]. In *N. crassa*, the negative regulatory factor PREG can control the activity of transcription factor NUC-1, while NUC-1 can up-regulate the activity of the phosphorus acquisition enzymes encoding genes during phosphorus starvation conditions [62,115,116]. Interestingly, a homolog of *N. crassa* NUC was identified in *V. dahliae* as VdNUC-2, and is required for the pathogen’s full virulence [117]. The lower activity of PREG in the highly aggressive *V. dahliae* isolate compared to the weakly aggressive one may indicate a higher activity of the phosphorus acquisition pathway. Phosphate acquisition is important for virulence in *Ustilago maydis* and *C. neoformans* and a connection was shown between phosphate acquisition and the PKA pathway controls the fungal virulence in both pathogens [118,119]. Our findings in PREG are consistent with their research as well as our hypothesis that a highly aggressive *V. dahliae* may suppress the up-regulation of PREG that is occurring in weakly aggressive *V. dahliae*, in order to achieve a higher activity in NUC1 and the phosphate acquisition pathway, as well as the serine/threonine protein kinases (such as PKA).

Finally, HAD-like hydrolase protein can be identified in different organisms, and studies reported their important roles in virulence in *C. neoformans* and in detoxification in *E. coli* and in *S. cerevisiae* [66,67,120]. Plants produce various secondary metabolites, proteases, and phytoalexins to ward off pathogens [121,122,123,124,125]. Plants also release some endogenous peptides to induce plant defense [126]. HAD superfamily hydrolase might help *V. dahliae* avoid plant toxins by detoxification and therefore play a role in virulence. This research is in line with our findings that HAD superfamily hydrolase may have a role to play in infection and differential pathogenic abilities of *V. dahliae* isolates.

## 4. Materials and Methods

### 4.1. V. dahliae Isolates

Two *V. dahliae* isolates, Vs06-07 and Vd1396-9, a weakly and highly aggressive isolate, respectively [127,128], were selected for this study. Both isolates were cultured on potato dextrose agar (PDA) 24 ± 0.5 °C for 14 days. 

### 4.2. Pathogenicity Tests of V. dahliae Isolates

Kennebec, a susceptible potato cultivar to *V. dahliae* [127], was used for the detached leaves inoculation and pathogenicity tests. Plants were grown in a mixture of sand, soil, and peat moss with a ratio of 16:4:1, under a day/light temperature regimen of 22/18 °C and a photoperiod of 16/8 h. 

Assessment of the pathogenicity of both *V. dahliae* isolates was performed on potato cv. Kennebec as described by Zhu, Soliman [24]. Briefly, three Kennebec plants (3-week-old) were trimmed at the root tips and placed in the conidia suspension (10^6^ conidia/mL) of the weakly or highly aggressive *V. dahliae* isolates, then planted in 6-inch pots with a pasteurized mixture of sand, soil, and peat moss (16:4:1). The total area under disease progress curves (AUDPC) of disease severity and percentage of infection were evaluated as described by Zhu, Soliman [24].

### 4.3. Gene Selection and Primers’ Design

Fifteen genes were selected based on proteomic and subtractive hybridization/cDNA-AFLP analyses of weakly and highly aggressive isolates in previous work [22,23]. These include: the genes encoding Ras-GAP like protein (VDAG_01012), pyruvate dehydrogenase E1 component subunit beta (VdPDHB) (VDAG_01642), DNA-(apurinic or apyrimidinic site) lyase (DNA AP lyase) (VDAG_02445), glucan endo-1,3-alpha-glucosidase agn1 (VDAG_04101), thioredoxin (VdTRX) (VDAG_04529), serine/threonine protein kinase (VDAG_04632), ubiquitin-conjugating enzyme variant MMS2 (VDAG_05365), nuc-1 negative regulatory protein preg (VdPREG) (VDAG_06766), xanthine dehydrogenase (VDAG_07735), myo-inositol 2-dehydrogenase (VDAG_08205), HAD-superfamily hydrolase (VDAG_08490), DNA repair protein RAD51 (VDAG_08796), Wos2 (VDAG_08865), NADH-ubiquinone oxidoreductase (VDAG_09026), and serine 3-dehydrogenase (VDAG_09532). Primer pairs for the target genes were designed based on the sequences from Verticillium comparative genomic project available at the Broad Institute (https://www.broadinstitute.org/scientific-community/science/projects/fungal-genome-initiative/verticillium-comparative-genomics-pro (accessed on 7 March 2021)). The Histone H3 gene (VDAG_10035) was used as a housekeeping gene [24] (Table 1).

### 4.4. Elicitation and Inoculation Methods

The elicitation of both *V. dahliae* isolates with potato extracts was done following the protocol described by Zhu, Soliman [24], and the preparation of potato extracts was done following the protocol described by El-Bebany, Henriquez [23]. Briefly, one milliliter of 108 conidia of both isolates were added to 100 ml Czapek-Dox Broth (CDB) liquid media (Difco Laboratories, Sparks, MD, USA) and incubated at 24 ± 0.5 °C for 7 days. One milliliter of prepared potato leaf, stem, and root extracts was added into CDB medium, and mycelium samples were harvested after 7 days of incubation in the same conditions. Three biological replicates were prepared for each treatment. 

The detached leaves infection was done using a protocol described by Zhu, Soliman [24]. *V. dahliae* isolates Vd1396-9 and Vs06-07 were grown in PDA at 24 ± 0.5 °C for 21 days. Conidia were then collected and diluted in a concentration of 3 × 10^7^ conidia/mL. Clones of Potato cv. Kennebec plants were then grown in LA4 soil mix (SunGro Horticulture, Agawam, MA 01001, USA) for 4 weeks, then one leaf was cut, placed into 1 mL of the conidia suspension, and maintained under a photoperiod of 16/8 h at 24 °C. Sterilized water, was used as a control for inoculation. Four to six individual detached leaves, each from separate plants were pooled as one sample. This was repeated three times for each treatment at each time-point (1, 3, 5, and 8 days after inoculation (DAI)). Samples representing the three biological replicates were immediately frozen in liquid nitrogen and stored at −80°C until used for RNA extraction. 

### 4.5. RNA Extraction and RT-qPCR

Total RNA was extracted from mycelium (100 mg), and potato detached leaves (100 mg) using the Omega Fungal RNA kit (Omega Moi-Tek, Inc., Norcross, GA, USA), following the manufacturer’s protocol. The first strand cDNA was synthesized from 5 µg total RNA following the manufacturer’s protocol of the Superscript first strand synthesis kit (Life Technologies, Carlsbad, CA, USA).

Quantitative Real-Time PCR was performed for each target gene as well as for the reference gene using SsoFast EvaGreen Super mix (Bio-Rad Lab, Hercules, PA, USA) following the manufacturer recommendations. The 2^−ΔΔCT^ method [129] has been used to evaluate expression of the above mentioned genes. 

### 4.6. Statistical Analysis

Pathogenicity data and gene expression under different treatments were analyzed with PROC MIXED in SAS Statistical Analysis Software (SAS Institute, Cary, NC, USA; release 9.1 for Windows). The normality of all data (three biological replicates, in infection experiments each representing 4–6 leaves samples from 4–6 separate plants) was determined with PROC UNIVARIATE, and analysis showed that data from different treatments were all qualified for normal distribution with Shapiro–Wilk test (>0.9) and p value (>0.05), and qualified for homogeneity based on residuals comparison to studentized residuals critical values [130]. Log10 transformation was applied to some sets of gene expression data statistical analysis. The macro PDMIX800.sas [53], α = 0.05 was applied to mean values separated by least squared means, and into grouped letter results. Mean values (n = 3) with the same letters indicate non-significant differences from each other (*p* < 0.05) in different sets of experiments.

## 5. Conclusions

In conclusion, the expression level of most of the genes we tested increased in both weakly and highly aggressive isolates in response to treatment with the various potato extracts. The metabolism of pyruvate, purines and pyrimidines, inositol and inositol phosphate, and phosphorus acquisition may be activated more in the highly aggressive isolate than in the weakly aggressive one in response to the various potato extracts (Figure 7). The genes involved in ROS production (NADH-ubiquinone oxidoreductase) and small GTPase signaling pathway (including Ras-GAP and serine/threonine protein kinase) were also elevated in the highly aggressive isolate more than the weakly aggressive one when exposed to potato stem extracts, suggesting a higher activity in these pathways (Figure 7). During infection of potato detached leaves, there were more up-regulated genes in the highly aggressive isolate than the weak aggressive one. These genes include key elements in the small GTPase signaling pathway, which include Ras-GAP and serine/threonine protein kinase, DNA damage repair, ROS production and cleavage, cellular metabolism, and detoxification. These results suggest an important role for all of these pathways during infection. This study provides a detailed follow up on the temporal expression of genes identified in our previous studies [22,23]. Together with more recent functional analyses of selected genes and interaction with host defense components [21,24,131], this study provides another background to reflect on which metabolic reactions contribute to disease, and which ones rather contribute to less disease or lack of it. The work ahead should put emphasis on the spatiotemporal expression of these genes during these interactions in order to fine-tune the expression map of genes that are important for the pathogen to invade its hosts. Further functional analyses of these genes should validate their involvement in an integrated picture of the arsenal that *V. dahliae* uses against its most economic hosts. This will no doubt help us to the right direction of future management tools against Verticillium wilt disease. 

## Figures and Tables

**Figure 1 pathogens-10-00510-f001:**
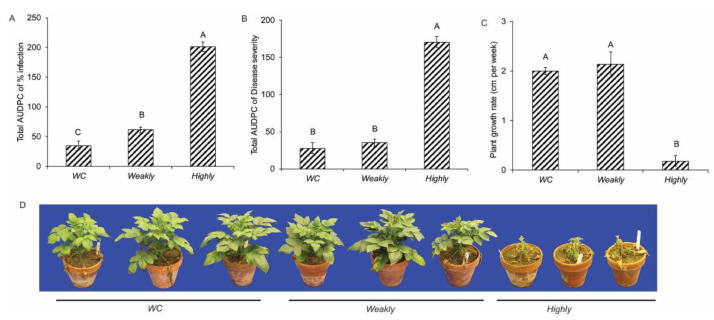
Pathogenicity analysis of the highly (Vd1396-9) and weakly aggressive (Vs06-07) *V. dahliae* isolates on the susceptible potato cultivar Kennebec. (**A**) Total AUDPC of percent infection; (**B**) total AUDPC of disease severity; (**C**) growth rate of potato plants per week (cm); (**D**) potato plants infected with WC, Weakly and Highly aggressive isolates at 5 weeks after inoculation. WC: Water control treatment; Weakly: weakly aggressive *V. dahliae* isolate Vs06-07; Highly: highly aggressive *V. dahliae* isolate Vd1396-9. Error bars refer to standard errors. The macro PDMIX800.sas [53], α = 0.05 was applied to mean values separated by least squared means, and into grouped letters’ results. Values (n = 3) with the same letters are not significantly different (*p* < 0.05).

**Figure 2 pathogens-10-00510-f002:**
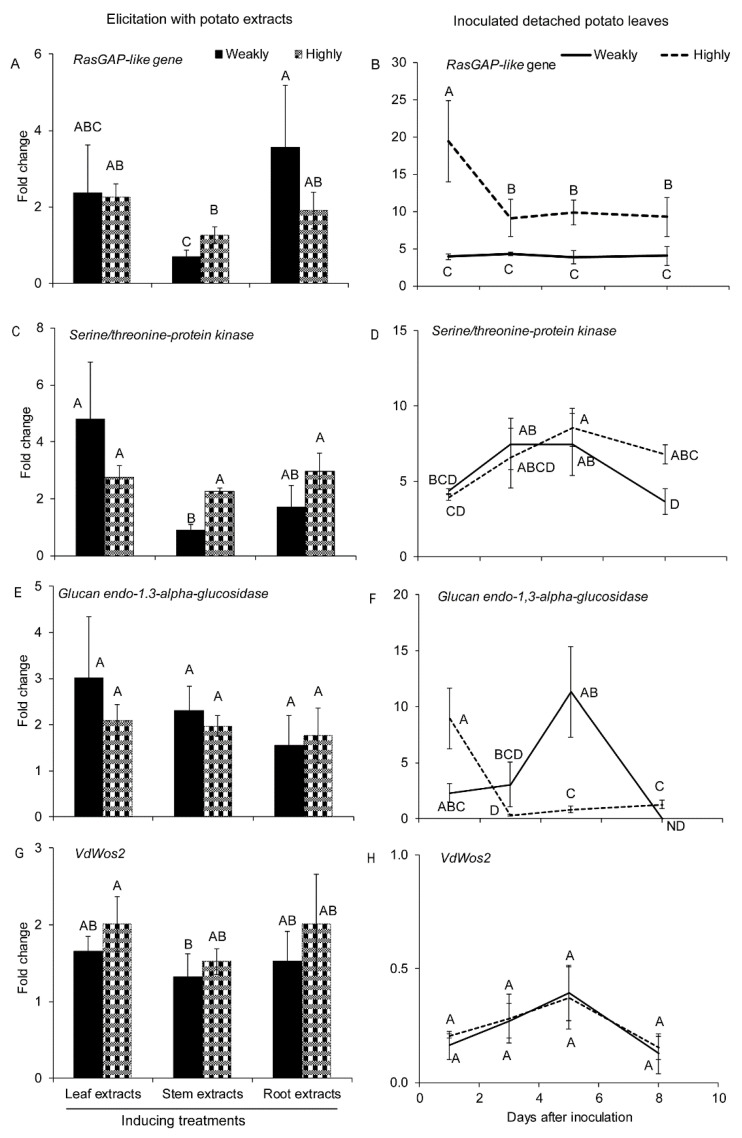
Expression of genes involved in cell differential and proliferation under different treatments. Expression of (**A**) Ras-GAP like protein, (**C**) serine/threonine protein kinases, (**E**) glucan endo-1,3-alpha-glucosidase and (**G**) VdWos2 in the highly (Vd1396-9) and the weakly (Vs06-07) aggressive *V. dahliae* isolates in response to elicitation with different potato extracts: leaf, stem, root extracts. Expression of (**B**) Ras-GAP like protein, (**D**) serine/threonine protein kinases, (**F**) glucan endo-1,3-alpha-glucosidase and (**H**) VdWos2 in the highly (Vd1396-9) and the weakly (Vs06-07) aggressive *V. dahliae* isolates in response to infection of detached potato leaves at 1, 3, 5, and 8 DAI. WC: Water control treatment; Weakly: weakly aggressive *V. dahliae* isolate Vs06-07; Highly: highly aggressive *V. dahliae* isolate Vd1396-9; ND: Non-detectable. Error bars refer to standard error. The macro PDMIX800.sas [53], α = 0.05 was applied to mean values separated by least squared means, and into grouped letters’ results. Values (n = 3) with the same letters are not significantly different (*p* < 0.05).

**Figure 3 pathogens-10-00510-f003:**
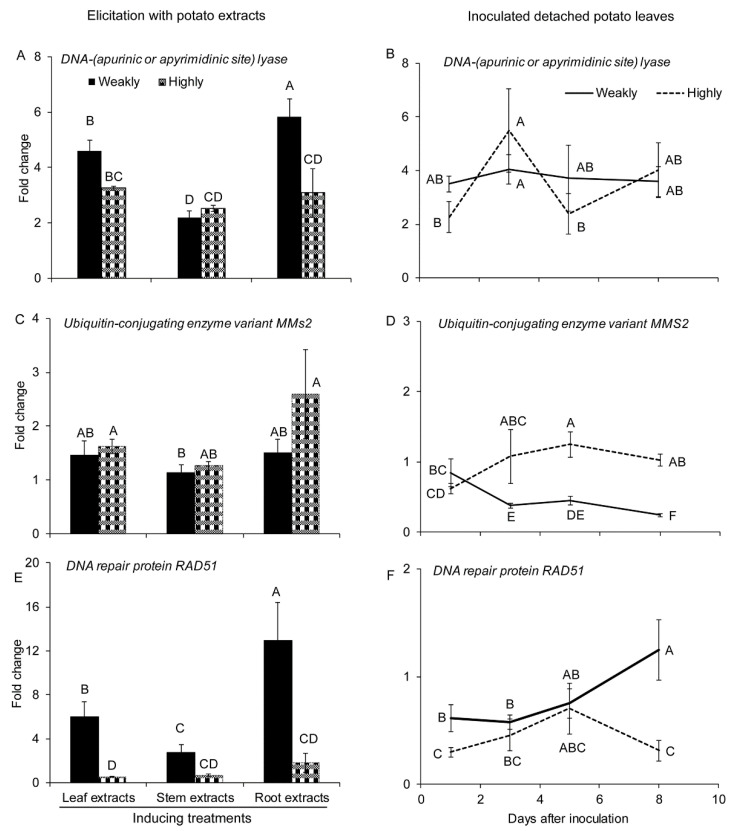
Expression of genes involved in DNA repair under different treatments. Expression of (**A**) DNA-(apurinic or apyrimidinic site) lyase, (**C**) ubiquitin-conjugating enzyme variant MMS2, and (**E**) DNA repair protein RAD51 in the highly (Vd1396-9) and the weakly (Vs06-07) aggressive *V. dahliae* isolates in response to elicitation with different potato extracts: leaf, stem, root extracts. Expression of (**B**) DNA-(apurinic or apyrimidinic site) lyase, (**D**) ubiquitin-conjugating enzyme variant MMS2, and (**F**) DNA repair protein RAD51 in the highly (Vd1396-9) and the weakly (Vs06-07) aggressive *V. dahliae* isolates in response to infection of detached potato leaves at 1, 3, 5, and 8 DAI. WC: Water control treatment; Weakly: weakly aggressive *V. dahliae* isolate Vs06-07; Highly: highly aggressive *V. dahliae* isolate Vd1396-9. Error bars refer to standard error. The macro PDMIX800.sas [53], α = 0.05 was applied to mean values separated by least squared means, and into grouped letters results. Values (n = 3) with the same letters are not significantly different (*p* < 0.05).

**Figure 4 pathogens-10-00510-f004:**
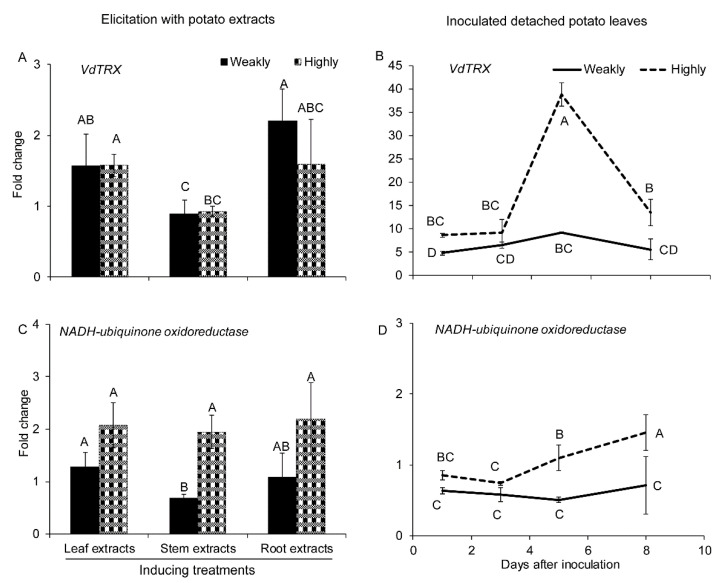
Expression of genes involved in ROS regulation under different treatments. Expression of (**A**) Thioredoxin (Trx) and (**C**) NADH-ubiquinone oxidoreductase in the highly (Vd1396-9) and the weakly (Vs06-07) aggressive *V. dahliae* isolates in response to elicitation with different potato extracts: leaf, stem, root extracts. Expression of (**B**) Trx and (**D**) NADH-ubiquinone oxidoreductase in the highly (Vd1396-9) and the weakly (Vs06-07) aggressive *V. dahliae* isolates in response to infection of detached potato leaves at 1, 3, 5, and 8 DAI. WC: Water control treatment; Weakly: weakly aggressive *V. dahliae* isolate Vs06-07; Highly: highly aggressive *V. dahliae* isolate Vd1396-9. Error bars refer to standard error. The macro PDMIX800.sas [53], α = 0.05 was applied to mean values separated by least squared means, and into grouped letters results. Values (n = 3) with the same letters are not significantly different (*p* < 0.05).

**Figure 5 pathogens-10-00510-f005:**
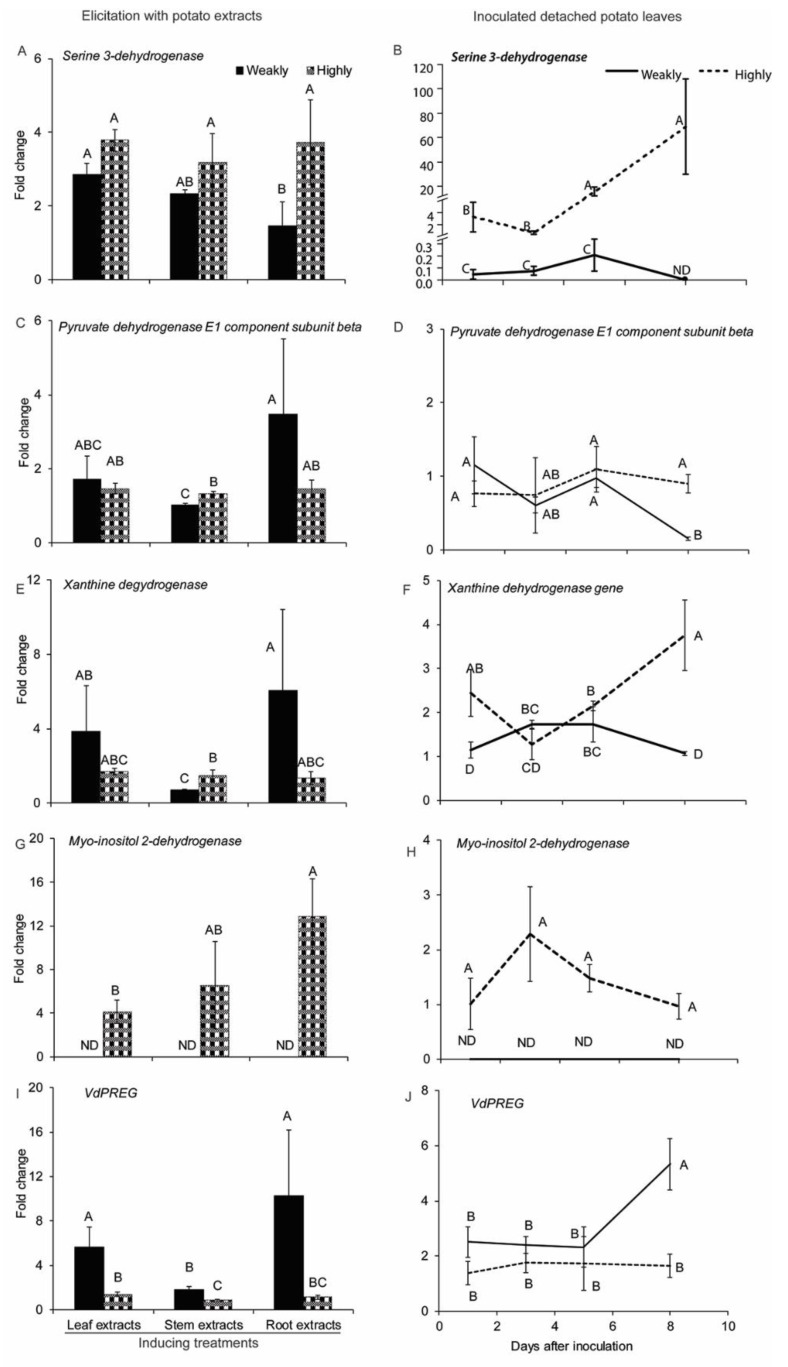
Expression of genes related to cellular metabolism under different treatments. Expression of (**A**) Serine 3-dehydrogenase, (**C**) Pyruvate dehydrogenase E1 component subunit beta, (**E**) Xanthine dehydrogenase, (**G**) Myo-inositol dehydrogenase, and (**I**) *VdPREG* in the highly (Vd1396-9) and the weakly (Vs06-07) aggressive *V. dahliae* isolates in response to elicitation with different potato extracts: leaf, stem, root extracts. Expression of (**B**) Serine 3-dehydrogenase, (**D**) Pyruvate dehydrogenase E1 component subunit beta, (**F**) Xanthine dehydrogenase, (**H**) Myo-inositol dehydrogenase, and (**J**) *VdPREG* in the highly (Vd1396-9) and the weakly (Vs06-07) aggressive *V. dahliae* isolates in response to infection of detached potato leaves at 1, 3, 5, and 8 DAI. WC: Water control treatment; Weakly: weakly aggressive *V. dahliae* isolate Vs06-07; Highly: highly aggressive *V. dahliae* isolate Vd1396-9; ND: Non-detectable. Error bars refer to standard error. The macro PDMIX800.sas [53], α = 0.05 was applied to mean values separated by least squared means, and into grouped letters’ results. Values (n = 3) with the same letters are not significantly different (*p* < 0.05).

**Figure 6 pathogens-10-00510-f006:**
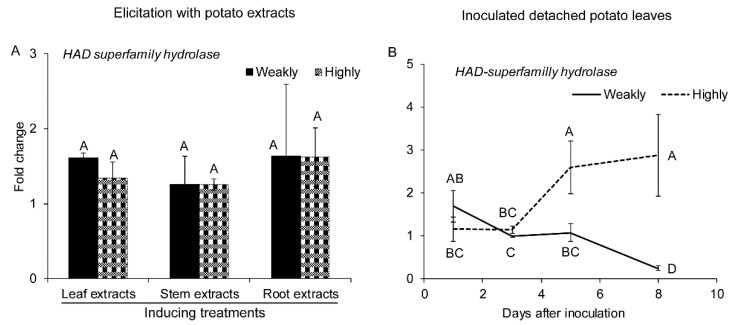
Expression of genes related to detoxification under different treatments. (**A**) Expression of HAD superfamily hydrolase in the highly (Vd1396-9) and the weakly (Vs06-07) aggressive *V. dahliae* isolates in response to elicitation with different potato extracts: leaf, stem, root extracts. (**B**) Expression of HAD superfamily hydrolase in the highly (Vd1396-9) and the weakly (Vs06-07) aggressive *V. dahliae* isolates in response to infection of detached potato leaves at 1, 3, 5, and 8 DAI. WC: Water control treatment; Weakly: weakly aggressive *V. dahliae* isolate Vs06-07; Highly: highly aggressive *V. dahliae* isolate Vd1396-9. Error bars refer to standard error. The macro PDMIX800.sas [53], α = 0.05 was applied to mean values separated by least squared means, and into grouped letters results. Values (n = 3) with the same letters are not significantly different (*p* < 0.05).

**Figure 7 pathogens-10-00510-f007:**
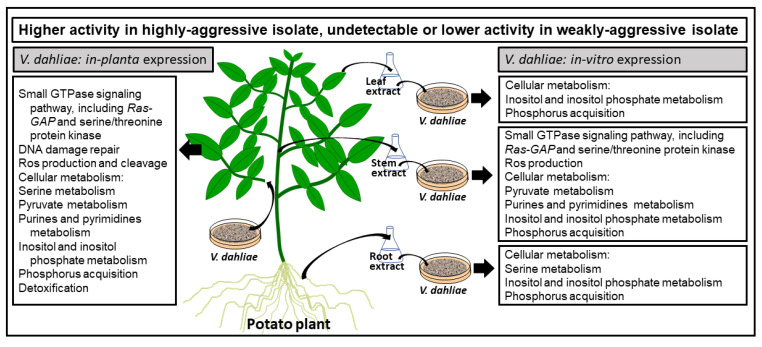
Potential involvement of pathways in response to potato extracts from various tissues and infection.

**Table 1 pathogens-10-00510-t001:** Primers used in RT-qPCR for amplifying the target genes.

Primer’s Name	Primer Sequence	Tm (°C)	Accession Number	Amplification Length (bp)
Ras-GAP like protein-F	ACGCTGTCCAACCTTCAC	53	VDAG_01012	273
Ras-GAP like protein-R	GTTGATCTTGTCCCAGTCG	52.7	VDAG_01012	
Pyruvate dehydrogenase E1 component subunit beta-F	CTTCGGCGACAAGAGGGT	58.1	VDAG_01642	283
Pyruvate dehydrogenase E1 component subunit beta-R	GGGAATGCTGCCATACCAC	58.2	VDAG_01642	
DNA-(apurinic or apyrimidinic site) lyase-F	CCGGCTGGGACGTTTA	55.1	VDAG_02445	152
DNA-(apurinic or apyrimidinic site) lyase-R	GCGGAATCTGGTGGTTG	54.2	VDAG_02445	
glucan endo-1,3-alpha-glucosidase agn1-F	GCCTTCGGAAACCTCAAT	54.6	VDAG_04101	307
glucan endo-1,3-alpha-glucosidase agn1-R	CTCCCATGAACTCATACGC	53.2	VDAG_04101	
Thioredoxin (VdTRX)-F	GCTGCTCCTGTTTATGCCTTTCCA	66.8	VDAG_04529	145
Thioredoxin (VdTRX)-R	GAGGTTATGCGGCTTGTTCGT	61.8	VDAG_04529	
serine/threonine protein kinase-F	GGTGGGTGCGGTCAAATA	56.9	VDAG_04632	272
serine/threonine protein kinase-R	AGGCATCCGTAGCACGAC	56.2	VDAG_04632	
Ubiquitin-conjugating enzyme variantMMS2-F	CATCCTCGGTCCTCCTCA	55.5	VDAG_05365	235
Ubiquitin-conjugating enzyme variantMMS2-R	CGCCATGTACCTCCTGATC	55.6	VDAG_05365	
VdPREG-F	GGGAATCTGACTAGGTTTCATT	54.7	VDAG_06766	227
VdPREG-R	GAGTCGGACAGACCTTTGG	54.7	VDAG_06766	
Xanthine dehydrogenase-F	GGCTGCTGCATGGATAAG	54.4	VDAG_07735	227
Xanthine dehydrogenase-R	CCGACAAATACCGACACG	54.9	VDAG_07735	
myo-inositol 2-dehydrogenase-F	AGTCTGGCATCGACAATAAC	52.3	VDAG_08205	192
myo-inositol 2-dehydrogenase-R	GCAGTCTCAACACGCAAA	52.6	VDAG_08205	
HAD-superfamily hydrolase-F	AGCCCGACCCTGCCATCTA	62.6	VDAG_08490	225
HAD-superfamily hydrolase-R	GGAACTCTTGCCAGTCCTTCA	59.2	VDAG_08490	
DNA repair protein RAD51-F	ATGGTGAGGGCGAGATGG	58.3	VDAG_08796	153
DNA repair protein RAD51-R	GGGTGTAAGCGACGGATT	55.4	VDAG_08796	
VdWos2-F	GTCTGCTACCAAGGCAACTCC	58.7	VDAG_08865	227
VdWos2-R	TCTCCTCCGTGTCAATCTCG	58.1	VDAG_08865	
NADH-ubiquinone oxidoreductase-F	ATCGGGGCGGGTCTCATT	62.1	VDAG_09026	116
NADH-ubiquinone oxidoreductase-R	CCTTCGGCAGGCTTCTCC	60.3	VDAG_09026	
Serine 3-dehydrogenase-F	ACTTGGGCATTAAGGTGGTC	56.5	VDAG_09532	348
Serine 3-dehydrogenase-R	CATCGCAGTCAGTTGTCGTAG	56.6	VDAG_09532	
His3-F (Zhu et al., 2017)	ATGGCTCGCACTAAGCAA	54.8	VDAG_10035	238
His3-R (Zhu et al., 2017)	TGAAGTCCTGGGCAATCT	52.7	VDAG_10035	

Note: The accession numbers of the genes are available in the Verticillium comparative genomic project of the Broad Institute (https://www.broadinstitute.org/scientific-community/science/projects/fungal-genome-initiative/verticillium-comparative-genomics-pro, accessed on 7 March 2021).

## Data Availability

The accession numbers of the genes are available in the Verticillium comparative genomic project of the Broad Institute (https://www.broadinstitute.org/scientific-community/science/projects/fungal-genome-initiative/verticillium-comparative-genomics-pro, accessed on 7 March 2021).

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
