# Peer review of "Gene Expression of Putative Pathogenicity-Related Genes in Verticillium dahliae in Response to Elicitation with Potato Extracts and during Infection Using Quantitative Real-Time PCR"

_pathogens, 2021, doi:10.3390/pathogens10050510_

Round 1

Reviewer 1 Report

Overall comments

The manuscript is well written, and the results are interesting, although limited without the inclusion of functional data. The fold changes and time course are interesting experiments that will add significantly to the literature. More overall context would be helpful, specifically in the introduction.

Introduction

Need more information as to why these specific genes were chosen. How many of these genes have signal peptides [running the amino acid sequence through SignalP (https://services.healthtech.dtu.dk/service.php?SignalP)]? Using the FASTA file developed by SignalP you can process the reads through EffectorP to determine if they are putative effectors (http://effectorp.csiro.au/). A broader literature review on the phylogenetic origins or any other background information on highly aggressive V. dahliae lines should be included. Are there any connections to the newly discovered race 2/3 in tomato (Usami et al. 2017; Chavarro-Carrero et al. 2020)? Providing the context in which isolates are related to other isolates around the world would be greatly beneficial. How were race differences ruled out? Is ‘Kennebec’ race 1 resistant?

Usami, T., Momma, N., Kikuchi, S., Watanabe, H., Hayashi, A., Mizukawa, M., Yoshino, K. and Ohmori, Y., 2017. Race 2 of Verticillium dahliae infecting tomato in Japan can be split into two races with differential pathogenicity on resistant rootstocks. Plant Pathology66(2), pp.230-238.

Chavarro-Carrero, E., Vermeulen, J., Torres, D., Usami, T., Schouten, H., Bai, Y., Seidl, M. and Thomma, B., 2020. Comparative genomics of Verticillium dahliae isolates reveals the in planta-secreted effector protein recognized in V2 tomato plants. BioRxiv.

Line

74 - Space missing after period.

Materials and Methods

RNA extraction - roughly how long were samples stored for before RNA extraction?

Was there any attempt to standardize samples or compare quality of samples (Nanodrop etc.?).

Line 432 – Should the 6 in “106 conidia/mL” be superscripted or written as “10^6 conidia/mL”.

Line 471 – There appear to be extra spaces at “days. Conidia”.

Line 471 – “3x107” should have the 7 superscripted or written as “3x10^7.

Please double check the rest of the manuscript for similar mistakes.

Results

The results in this study are presented well. Avoid adverbs such as ‘much’, especially in the results section. Results are either significant or not significant, if you wish to express a large change consider using exact numbers/fold changes rather than adverbs in the results section.

Reviewer 2 Report

The worked presented  an analysis of the expression of 15 genes putatively involved in pathogenicity from highly and weakly aggressive V. dahliae isolates. The authors used on expression profiling of these genes to associate their putative roles to two phases of V. dahliae’s interaction with potatos

The work is well presented and structured, and the methodology is well used but profiling gene expressions were only on base  mimics of both  pre-infection and infection from host.

I suggest the authors to consider the following points, as it could increase the quality of the manuscript:  

  • Assessing expression in vivo of some these genes during the phase infection would validate the data gene expression from mimic phases, giving further soporte to data and their hypothesis.
  • There are interesting information about Verticillium dahliae´s signal pathways, which would be useful in discussion.

Round 2

Reviewer 2 Report

In the Discussion, You had already included some information on
signalling pathways from other fungus that are related to your results but there are not information on Verticillium dalhiae´s signalling pathways related to your results. So I am still thinking  you should argue your results about signalling pathways using information from Verticillium. 

Author Response

Response to Reviewers’ Comments

Please note that line numbers referred to in the reviewers’ comments (boldface) represent those from the originally submitted manuscript, the line numbers referred to in our responses represent those found in the newly submitted manuscript.

Reviewer 2:

In the Discussion, You had already included some information on signalling pathways from other fungus that are related to your results but there are not information on Verticillium dalhiae´s signalling pathways related to your results. So I am still thinking you should argue your results about signalling pathways using information from Verticillium.

Authors’ response:

Thank you for pointing out this oversight in our Discussion. We now have added relevant information and references about Verticillium signalling pathways that are related to our results. Please see highlighted text (in blue) in Lines 342-345, Lines 355-357, Lines 365-371, Lines 399-402, Lines 424-431, and Lines 460-461.